# Redefining Radiation Metrics: Evaluating Actual Doses in Computed Tomography Scans

**DOI:** 10.3390/biomedicines12030600

**Published:** 2024-03-07

**Authors:** Dominika Sabiniewicz-Ziajka, Arkadiusz Szarmach, Małgorzata Grzywińska, Paweł Gać, Maciej Piskunowicz

**Affiliations:** 12nd Department of Radiology, Medical University of Gdansk, 80-210 Gdansk, Poland; dominika.sabiniewicz@gumed.edu.pl; 2Neuroinformatics and Artificial Intelligence Laboratory, Department of Neurophysiology, Neuropsychology and Neuroinformatics, Medical University of Gdansk, 80-210 Gdansk, Poland; malgorzata.grzywinska@gumed.edu.pl; 3Centre for Diagnostic Imaging, 4th Military Hospital, Weigla 5, 50-981 Wroclaw, Poland; pawelgac@interia.pl; 4Department of Population Health, Division of Environmental Health and Occupational Medicine, Wroclaw Medical University, Mikulicza-Radeckiego 7, 50-368 Wroclaw, Poland; 51st Department of Radiology, Medical University of Gdansk, 80-210 Gdansk, Poland; maciej.piskunowicz@gumed.edu.pl

**Keywords:** radiation dose, computed tomography, dose-length product, size-specific dose-length product, size-specific dose estimate

## Abstract

Background: Computed tomography (CT) contributes significantly to the collective dose from medical sources, raising concerns about potential health risks. However, existing radiation dose estimation tools, such as volume computed tomography dose index (CTDIvol), dose-length product (DLP), effective dose (ED), and size-specific dose estimate (SSDE), have limitations in accurately reflecting patient exposure. This study introduces a new parameter, size-specific dose-length product (DLPss), aiming to enhance the precision of radiation dose estimation in real-life scenarios. Methods: A retrospective analysis of 134 chest CT studies was conducted. Relationships between CTDIvol and anthropometric parameters were examined, and SSDE was calculated based on effective diameter. Additionally, the novel parameter, DLPss, was introduced, considering scan length and cross-sectional dimensions. Results: Analysis reveals variations in scan length, effective diameter, and CTDIvol between genders. Strong correlations were observed between CTDIvol and effective diameter, particularly in men. The average CTDIvol for the entire group was 7.83 ± 2.92 mGy, with statistically significant differences between women (7.38 ± 3.23 mGy) and men (8.30 ± 2.49 mGy). SSDE values showed significant gender differences, with men exhibiting higher values. The average SSDE values for women and men were 9.15 ± 2.5 mGy and 9.6 ± 2.09 mGy, respectively, with a statistically significant difference (*p* = 0.03). The newly introduced DLPss values ranged around 343.90 ± 81.66 mGy·cm for the entire group, with statistically significant differences between women (323.53 ± 78.69 mGy·cm) and men (364.89 ± 79.87 mGy·cm) (*p* < 0.05), providing a comprehensive assessment of total radiation dose. Conclusion: The study highlights the need for accurate radiation dose estimation, emphasizing the impact of CT examination parameters on dose variability. The proposed DLPss parameter offers a promising approach to enhancing precision in assessing radiation risk during CT scans. Further research is warranted to explore additional parameters for a comprehensive understanding of radiation exposure and to optimize imaging protocols for patient safety.

## 1. Introduction

Since the introduction of multi-row-detector computed tomography (MDCT) in the 1990s, there has been a steady increase in the number of imaging studies using this technique [1,2,3,4]. It is estimated that approximately 400 million tomographic examinations are performed worldwide each year [5], and over the past 10 years, their quantity has increased by 82% [5]. According to the United Nations Scientific Committee on the Effects of Atomic Radiation (UNSCEAR) report from 2020/21, computed tomography (CT) examinations accounted for 9.6% of all imaging studies, but they contribute to over 62% of the collective dose [5]. For comparison, conventional radiology (excluding dental) constitutes about 63% of radiological procedures but only contributes to 23% of the collective dose [5].

Due to the necessity of sometimes repeating CT scans, the actual doses received by patients accumulate and can often approach levels where the risk of negative effects from radiation exposure significantly increases. The Committee to Assess Health Risks from Exposure to Low Levels of Ionizing Radiation, in its Biologic Effects of Ionizing Radiation VII (BEIR VII) report, predicts that for every 100,000 individuals who receive a dose of ionizing radiation at the level of 100 mGy, up to 20,310 will die from cancer [6].

The approximate estimation of radiation dose received by a patient during a CT examination depends on the adopted methods of calculation and the applied correction parameters. The most direct way to estimate doses for patients undergoing CT examinations is by measuring the doses received by organs using phantoms with dimensions analogous to those of the patient [7]. Another method to obtain information about radiation dose involves complex mathematical calculations based on large amounts of data, utilizing analyses with virtual phantoms [8,9,10]. Unfortunately, these methods are subject to error stemming from the assumption that the patient being examined has dimensions comparable to those of the phantoms used, or a Monte-Carlo-based method to estimate radiation dose. If patients differ in size and body composition, it may be necessary to apply appropriate correction factors.

The methods of estimating real radiation dose still raise many questions. Available tools such as the volume computed tomography dose index (CTDIvol), dose-length product (DLP), effective dose (ED), or size-specific dose estimate (SSDE) do not fully describe the actual radiation dose to which a patient is exposed during scanning [11]. The research conducted aimed to analyze and identify factors influencing the dosage of ionizing radiation patients receive during imaging tests performed using computed tomography. Moreover, we have proposed a new parameter called size-specific dose-length product (DLPss), which may prove useful in more accurately estimating radiation dose in real-world scenarios.

## 2. Materials and Methods

This retrospective, cross-sectional, single-center study was approved by the Institutional Ethics Committee. The dedicated search engine MedStream Designer (Transition Technologies Science Ltd, Warsaw, Poland, 2015) was used to search the local database for examinations performed in the years 2020–2022. The sample population included computed tomography examinations of the chest in the adult population. Exclusion criteria consisted of patients with significant obesity (BMI > 35) or who were underweight (BMI < 18), previous chest surgery, chest deformities, fluid in the pleural cavities, pneumothorax, pneumonia, severe emphysema, and advanced lung cancer. Individuals under 18 years of age, low-dose lung examinations, low-quality studies with motion artifacts, and studies covering the chest, abdomen, and pelvis simultaneously were also excluded from the analysis. In the case of contrast-enhanced studies, only the examination in the native phase was assessed. The final analysis included a total of 134 patients: 68 women and 66 men (Table 1).

Every single one of the CT examinations was performed using the SOMATOM Definition Flash scanner (Siemens, Erlangen, Germany). All data were stored in the PACS archiving system and processed using the Syngo.via v.40 workstation (Siemens, Erlangen, Germany). Data were obtained for each X-ray radiation dose report regarding voltage, tube current, the diameter of the used standard phantom, volume computed tomography dose index (CTDIvol), dose-length product (DLP), topogram length, and actual scanning range. The anterior–posterior (AP) and lateral (LAT) dimensions were manually determined using an electronic ruler from the most extensive cross-sectional scan. The effective diameter (D eff) was calculated from the obtained data using a commonly accepted formula [12]:(1)$$\sqrt{AP\times LAT}$$

For the purposes of this study, we examined relationships between CTDIvol and selected anthropometric parameters (age, height, weight, BMI), as well as the length of the scan, AP and LAT dimensions, and effective diameter.

Based on data from The American Association of Physicists in Medicine Report No. 204 [13], the conversion factor (*k*) for each calculated effective diameter was read to estimate size-specific dose estimate (SSDE) according to the formula:**SSDE** = **CTDI_vol_** × ***k***(2)

To calculate the size-specific dose-length product (DLP) value for each patient, a parameter referred to as size-specific dose-length product (DLPss) has been introduced. It is represented as the product of DLP and the previously mentioned conversion factor *k*, which is determined based on the calculated effective diameter, as indicated in the formula:**DLP_ss_** = **DLP** × ***k***(3)

Meanwhile, the effective dose (ED) was calculated as the product of DLP and the protocol-specific dose conversion factor *ƒ*, according to the formula:**ED** = **DLP** × ***ƒ***(4)

For all chest CT examinations, based on AAPM Report No. 96, “The Measurement, Reporting, and Management of Radiation Dose in CT. Report of AAPM Task Group 23”, the value of the conversion factor *ƒ* was assumed to be 0.014 mSv/mGycm [14].

All calculations were performed using the statistical software TIBCO Software Inc. (Palo Alto, CA, USA, 2017) Statistica version 13. The Shapiro–Wilk test was applied for continuous variables to assess compliance with the normal distribution. Independent group comparisons based on gender and the examined anatomical region were conducted using the Student’s *t*-test for normally distributed variables and the Mann–Whitney U test for those deviating from normal distribution. Linear correlation coefficients were examined and calculated using non-parametric statistics with the Spearman rank correlation due to most variables’ lack of normal distribution. Analyses were performed with a 95% confidence interval, and a significance level of *p* < 0.05 was adopted for statistical hypothesis tests.

## 3. Results

### 3.1. The Scan Length

The average length of the topogram scan for the entire study group was 387.0 ± 48.04 mm (293–514 mm). Specifically, for women, it was 368.34 ± 38.29 mm (311–514 mm); for men, it was 406.20 ± 51.27 mm (293–514 mm). The proportions of topogram scan length to patient height were not significantly different between women and men, with values of 22.9% and 23.12%, respectively.

However, the length of the diagnostic scan varied between 321 and 476 mm (mean 400.33 ± 31.23 mm). These values exhibited statistically significant differences between women (386.18 ± 26.51 mm) and men (414.91 ± 29.08 mm). The scanned chest area to patient height ratio remained consistent for the entire group at 23.78%, with specific values of 24.0% for women and 23.63% for men.

In the analyzed studies, the lengths of diagnostic scans were, on average, 13.33 ± 45.76 mm longer than topogram scans. The percentage mean change in the length of the diagnostic scan compared to the topogram scan was 4.54 ± 11.35%, with changes ranging from −27.2% to +36.86%.

A detailed analysis of these changes revealed that in 29.1% of the studies, the diagnostic scan was shortened, while in 69.4%, it was extended compared to the topogram scan (Figure 1).

### 3.2. The Lateral (LAT) and Anterior–Posterior (AP) Dimensions and the Effective Diameter

In the analyzed group (without gender division), the average lateral (LAT) dimension was 35.07 ± 4.01 cm, the anterior–posterior (AP) dimension was 25.57 ± 3.38 cm, and the effective diameter was 29.92 ± 3.45 cm.

In the female group, these dimensions were as follows: lateral (LAT) dimension: 33.85 ± 4.18 cm, anterior–posterior (AP) dimension: 24.44 ± 3.12 cm, effective diameter: 28.74 ± 3.43 cm. In males, these dimensions were: lateral (LAT) dimension: 36.32 ± 3.44 cm, anterior–posterior (AP) dimension: 26.74 ± 3.26 cm, effective diameter: 31.13 ± 3.05 cm. It was demonstrated that in 69.9% of patients, the effective diameter was smaller than 32 cm (the diameter of the PMMA standard phantom [15]), while in 30.1% of the subjects, it was larger (Figure 2).

### 3.3. The Correlation of Effective Diameter with Body Mass Index

In the group of analyzed patients, a strong correlation between BMI and the effective diameter of the chest was demonstrated (r = 0.77) for the entire analyzed group, with r = 0.82 for women and r = 0.90 for men (Figure 3).

### 3.4. Volume Computed Tomography Dose Index (CTDI_vol_)

The averaged CTDIvol value for the entire group was 7.83 ± 2.92 mGy (range 3.2–18.12 mGy), with 7.38 ± 3.23 mGy (range 3.40–18.12 mGy) for women and 8.30 ± 2.49 mGy (range 3.2–14.53 mGy) for men. The differences between CTDIvol values for the female and male groups were statistically significant (*p* < 0.05)—higher in men.

### 3.5. Size-Specific Dose Estimate (SSDE)

The average SSDE value (Formula (2)) for the entire study group was 9.37 ± 2.31 mGy (range 4.2–17.05 mGy). Specifically, for women, the average was 9.15 ± 2.5 mGy (range 4.7–17.05 mGy), while for men, it was 9.6 ± 2.09 mGy (range 4.2–14.36 mGy). The observed differences were was statistically significant (*p* = 0.03). Notably, the calculated SSDE values were higher than the corresponding CTDI_vol_ values for the entire group by 20%.

### 3.6. The Relations between CTDIvol and Selected Anthropometric Parameters

For the purposes of this study, an analysis was conducted to investigate the correlation between CTDIvol and various anthropometric parameters of patients, including age, height, weight, BMI, scan length, AP and LAT dimensions, and effective diameter. Among women, the strongest correlation was identified with effective diameter (r = 0.79), whereas in men the strongest correlation was found with body weight (r = 0.88). The results are summarized in Table 2.

### 3.7. The Relations between Size-Specific Dose Estimate (SSDE) and Body Mass Index

A statistically significant relationship between SSDE and BMI was observed in both genders, with the correlation being significantly stronger in men (M/F r = 0.73/0.66) (see Figure 4a,b).

### 3.8. Dose Length Product (DLP)

Statistically significant differences in DLP values were observed between genders for both topographic and diagnostic scans. For men, these values were associated with a higher parameter value (refer to Table 3).

### 3.9. Size-Specific Dose-Length Product (DLP_ss_)

For the purpose of this analysis (see Section 2), a new parameter describing the total radiation dose received by an individual patient was introduced. This parameter takes takes into account both the actual scan length and the dimensions of its cross-section (DLPss). In the case of chest CT scans, the DLPss values in the studied group ranged around 343.90 ± 81.66 mGy·cm (151.8–558.6 mGy·cm). Specifically, for women, it was 323.53 ± 78.69 mGy·cm (169.9–523.45 mGy·cm), and for men, it was 364.89 ± 79.87 mGy·cm (151.8–558.6 mGy·cm), showing statistical significance (*p* < 0.05).

### 3.10. Effective Dose (ED)

The average values of effective dose (ED) for the entire group were 4.01 ± 1.4 mSv. However, for women, it was 3.62 ± 1.39 mSv, and for men, it was 4.41 ± 1.3 mSv.

## 4. Discussion

The continuous increase in the number of CT scans performed and the associated high effective dose make these examinations contribute significantly to radiation exposure [4,16,17,18,19,20]. It is estimated that the probability of stochastic radiation effects increases by 5% for every 1 Sv [21]. In practical terms, this implies that a single CT scan with an effective dose of 10 mSv increases the probability of such an effect by 0.05%. Translating these data to every 10,000 scans (each with a dose of 10 mSv), five patients will develop some form of cancer as a result of the adverse effects of ionizing radiation. Although the risk for an individual patient is not very high, the growing number of individuals undergoing diagnostic examinations contributes to an overall increase in the population’s absolute number of cancers resulting from radiation exposure [4,22,23].

Most computed tomography examinations, following the acquisition of the topographic scan, involve two phases: a non-contrast study and a phase with intravenous administration of contrast agent. However, it is important to remember that each phase of the study involves radiation dose multiplication. According to the findings of Smith-Bindman et al., the radiation dose for a multiphase chest CT scan was on average twice as high compared to a single-phase scan (18 mSv vs. 9 mSv) [24].

Current therapies, especially in oncology, require periodic monitoring of treatment effectiveness, typically using imaging studies involving ionizing radiation. These examinations significantly increase the cumulative dose, and, in extreme cases, they can lead to serious adverse effects due to direct exposure to ionizing radiation on the body [5,6]. As reported by Wiest et al., such a cumulative dose can reach 50 to even 100 mSv, which is similar to the dose received by individuals who survived the detonation of an atomic bomb [25].

In light of the above data, the development of reliable methods for calculating radiation dose and its precise monitoring becomes paramount. This will enable the implementation of measures actions aimed at reducing radiation risk, modifying examination protocols, and advancing successive generations of CT scanners.

The actual dose received by the patient (patient effective dose) depends on a number of factors, such as table settings, the applied examination protocol, the type of scanner, and the size and physique of the patient. The selection of the scanning area size is an effective means to limit radiation dose. Unfortunately, according to Campbell et al., up to 98% of chest CT scans exceed their planned range [26]. Additional scans do not provide significant diagnostic information; however, they contribute to an elevation in the radiation dose (DLP), ranging from 39.98 to 132.59 mGy·cm. Cohen et al. conducted an insightful analysis of the segment length examined in relation to anatomical boundaries and topographic scan length within a cohort of 1118 chest CT scans [27]. Their study revealed that the average length of the diagnostic scan extended 25 mm above the upper lung boundary (5 mm greater than the length of the topographic scan) and 49 mm below the diaphragm (29 mm greater than the length of the topographic scan). Notably, 81% of the analyzed scans exceeded the upper lung boundary, while 95% extended below it. The authors attribute these findings to challenges in accurately delineating the lower lung boundary, primarily stemming from respiratory motion, and recommend setting scanning boundaries with a 2 cm margin to mitigate such challenges. Additionally, Cohen and co-authors analyzed other factors contributing to unjustified extensions of the scanned area. Younger age and male gender of the patient inclined the technician to increase the scanning area. The authors also investigated the impact of the number of studies performed by technicians, and even the day of the week when studies with extended scans were most frequently performed. Interestingly, more experienced technicians more often extended scans length, which can be explained by excessive workload rather than routine. However, the day on which the most studies with an extended scanning range were performed was Friday (24%) [27]. Liao et al. analyzed the results of 442 CT scans of the chest and abdomen. Remarkably, 99% of them covered too large a body area. The average extension of diagnostic scans was 43.2 mm, but in extreme cases, it reached even 180 mm, resulting in an increase in radiation dose from 8.4 to 10.38% [28]. According to Zanca et al., the unjustified extension of scans, despite clinical indications, resulted in a significant increase in the effective dose for chest scans from 4.2 mSv to 4.8 mSv. It also exposed adjacent organs; in the mentioned study, an increase in the dose to the thyroid by 99% and to the breast by even 163% was observed [29].

In our study, the length of the scanned segment was reduced in almost 30% of the scans, while it was extended in nearly 70% compared to the length of the topographic scan (see Figure 1). This indicates that a significantly larger proportion of studies extended the diagnostic scan beyond the topogram, leading to an unnecessary increase in radiation dose.

It is well known that the dimensions of the cross-sectional area of the scanned region are closely correlated with the radiation dose. This correlation directly affects the volumetric computed tomography dose index (CTDIvol) and, consequently, the determination of the size-specific dose estimate (SSDE) conversion factor during dose estimation. In many publications, various methods for calculating the dimensions of the cross-sectional area and determining the effective diameter (Deff) can be found [30,31,32]. However, the key factor appears to be the choice of the appropriate cross-sectional plane. The most commonly used techniques measure the cross-section at the central point of the scanned area (Deff c) or the cross-section with the largest dimensions (Deff m). Due to the irregular shape of the examined regions, obvious differences in measurements arise. The most accurate and precise measurement seems to be the diameter measurement based on the cross-sectional area (Deff a). Unfortunately, calculating the diameter using this method is very labor- and time-consuming.

Anam et al. proposed an interesting solution to this problem [31]. They attempted to compare the effective diameter calculated based on anterior–posterior (AP) and lateral (LAT) measurements of the central cross-section (Deff c) and the maximum cross-section (Deff m), with the diameter calculated based on the cross-sectional area (Deff a) treated as the reference value. The effective diameter of the maximum cross-section (Deff m) did not show statistically significant differences compared to the diameter calculated from the cross-sectional area (Deff a). The percentage differences between measurements for chest CT were 2.0% and statistically insignificant (*p* > 0.05). However, for the central cross-section (Deff c), the effective diameters calculated from the cross-sectional area (Deff a) showed statistically significant differences, amounting to 5.5%. In light of the above, the authors of the publication do not recommend using measurements obtained from the central cross-section to calculate the effective diameter [31].

The newest parameter—the effective diameter ratio (Deff r)—was proposed by Lamoureux et al. as a supplement to data on patient size [33]. It is the ratio of the external diameter of the patient to the internal diameter (excluding subcutaneous fat tissue). It is intended to provide information about the anatomical composition of the body, especially the volume of subcutaneous fat tissue. However, it does not assess intra-abdominal obesity.

In our dataset, the mean effective diameter was determined to be 29.92 ± 3.45 cm, deviating by 6.5% from the reference phantom diameter of 32 cm. Furthermore, in nearly 70% of the studied patients, the effective chest diameter was smaller than the phantom diameter, while in approximately 30%, it exceeded this value (see Figure 2). This observation suggests that the radiation dose reported in the examination protocol, calculated using CTDIvol, may be underestimated in a significant subset of patients, emphasizing the necessity for size-specific dose estimation in computed tomography, depending on the individual patient’s size.

Many authors support the thesis that body mass can be a good indicator of the patient’s body size, making it a valuable tool for adjusting examination parameters [34,35]. However, Nyman et al. recommended the measurement of the patient’s body circumference [36]. It should be noted that areas of the same cross-sectional size or comparable circumference may exhibit varying degrees of radiation absorption due to differences in body composition. These relationships were investigated by McLaughlin et al. [37]. Their study demonstrated a strong correlation between radiation dose and the total volume of the patient’s adipose tissue (r = 0.77), which proved to be the strongest predictor of radiation dose, even stronger than BMI.

Boos et al. demonstrated a significant correlation between effective diameter (Deff) and BMI (r = 0.85), as well as between effective diameter and the patient’s body mass (r = 0.84). For chest imaging, effective diameter correlated better with body mass than BMI (r = 0.87 vs. r = 0.81) [38]. A similar relationship was presented in the study by Khawaja et al. [39].

In our study, the highest correlation was found between CTDIvol values and effective diameter: r = 0.79 in women and r = 0.87 in men. However, the correlation with BMI was less pronounced, with values of r = 0.74 and r = 0.86, respectively.

Efforts have been made to demonstrate that other anthropometric parameters may also correlate with the effective diameter. In the previously mentioned study [38], the authors also assessed the usefulness of patient age in estimating their effective diameter. In the adult population, no correlation was found between this parameter and patient age, and such a relationship was only observed in children.

Modern CT scanners equipped with attenuation-based tube current modulation (ATCM) systems modify changes in tube current intensity based on the individual patient’s anatomy. This system leads to a logarithmic increase in tube current intensity with the increase in patient dimensions [40]. Consequently, the radiation dose for obese patients is higher [41]. This results from both the automatic increase in tube current and the reduction in the distance between the patient and the tube [42]. Therefore, it is crucial to determine correction parameters that allow for estimating the actual dose received by a specific patient. For this reason, the size-specific dose estimate (SSDE) parameter was introduced, which is the product of CTDIvol and a conversion factor (*k*) derived from a series of anthropometric parameters, with the effective diameter of the patient proving to be the most useful [13,31,39,43]. In AAPM Report No. 220, the conversion factor is determined based on the diameter of a water cylinder that would attenuate radiation to the same extent as the patient’s body [44]. Unfortunately, this method necessitates the use of additional software. It is important to note that SSDE reflects the dose only for a single layer of the cross-sectional area. Consequently, measuring this parameter is associated with a methodological error and may exhibit variability in successive scans. Hence, automating the determination of SSDE for each cross-section and including these values in the report for each examination would be advisable.

For the purpose of this study, the correction factor used to calculate SSDE was determined based on the effective diameter of the maximum cross-sectional area. This resulted in higher radiation doses than those reported in the examination protocol for almost 70% of the subjects. Simultaneously, the average CTDIvol values were higher by almost 20%

The radiation dose parameter for the entire examination is the dose length product (DLP), which is the product of CTDIvol and the length of the scanned area. Unfortunately, the DLP value is close to the actual dose only when the diameter of the entire scanned area is similar to the diameter of the reference phantom. However, the effective diameter of the cross-sectional area of the patient is rarely equal to the phantom diameter.

In our study, the Deff value in nearly three-quarters of the examined patients was smaller, and in one-third larger than the diameter of the reference phantom.

Currently, CTDIvol and DLP are commonly used and legally approved parameters for assessing the radiation dose reported in each examination protocol. Nevertheless, both of these parameters have significant drawbacks.

Another parameter assessing the radiation risk to the patient is the effective dose (ED). This indicator is the product of DLP and the conversion factor ***ƒ***. Unfortunately, even this parameter is determined for the reference phantom, not for a specific patient. For this work, the value of the conversion factor *ƒ* = 0.014 mSv/mGy·cm was adopted from the AAPM report No. 96 [14].

In accordance with currently applicable guidelines, calculating the effective dose based on the dose-length product is considered reliable. The values of ED calculated using this parameter do not differ significantly from those obtained using more restrictive methods, and these differences range between 10% and 15% [14,45,46].

The previously mentioned study by Smith-Bindman et al. demonstrated that the average effective dose for chest CT scans was 9 mSv (range 5–13 mSv) [24]. Relatively low average ED values were obtained from the analysis of over 150,000 studies conducted in the Chinese province of Sichuan. Specifically, for chest CT scans, they were 5.1 mSv [47].

In our own study, the average values of effective dose (ED) were 4.01 ± 1.4 mSv for the entire group, 3.62 ± 1.39 mSv for women, and 4.41 ± 1.3 mSv for men.

As mentioned above, SSDE describes the radiation dose for a single cross-sectional layer. Therefore, applying this factor to calculate parameters for the actual radiation dose for the entire study, such as dose-length product and effective dose appears to be associated with a methodological error.

For the purposes of this study, a modification of DLP values has been proposed based on size-adjusted values for a specific patient. Utilizing a conversion factor calculated for the actual effective diameter (*k*), DLPss was determined according to the formula DLPss = DLP × *k*. The values determined for DLPss using this approach were found to be significantly higher than values calculated in the traditional manner, averaging 20% for both parameters.

The novel method proposed by us for dose calculation appears to more accurately reflect the actual radiation dose received during the scanning of a specific patient, thereby facilitating a more precise estimation of the radiation risk associated with computed tomography research protocols. This may result in a tangible reduction in radiation dose during various types of scanning.

Due to significant variations in radiation doses associated with the same scanning protocols in 1996, the idea of establishing diagnostic reference levels (DRL) emerged. The process of establishing DRL involved gathering radiation dose parameters from institutions in countries participating in this program [24,48,49,50,51]. The International Atomic Energy Agency (IAEA 2006) and the International Commission on Radiological Protection (ICRP 2007) recommended the establishment of diagnostic reference levels (DRL) to optimize radiation doses [52,53]. As reference benchmarks for individual radiation dose parameters, values at the 75th percentile are provided. These determined reference levels enable an objective assessment of radiation dose values, making them a valuable data source for comparing CT examination protocols across different countries, institutions, scanner types, and populations. Considering these factors could contribute to reducing disparities in reference doses.

A detailed analysis of the data revealed that significant differences in effective dose (ED) values between individual countries or CT facilities were predominantly attributable to variations in the settings of the scanner’s technical parameters and the applied protocols, rather than differences in the anthropometric parameters of the examined patients or the type and model of the scanner. While age and the specific CT scanner model had minor impacts, they only minimally contributed to the observed discrepancies in dose. Notably, it was observed that individual preferences of those performing and evaluating the examinations also significantly influenced radiation dose outcomes [53,54].

The increasing integration of artificial intelligence (AI) in patient radiological protection is conspicuous, evidenced by the development of dedicated algorithms tailored to automatically calculate the optimal radiation dose for individual patients. These algorithms consider various factors, including body size, age, and specific diagnostic requirements. Furthermore, deep learning techniques hold promise in enhancing image reconstruction, potentially mitigating the necessity for higher radiation doses to achieve diagnostically acceptable images. Notably, empirical evidence indicates that such automated solutions can result in a noteworthy reduction in radiation dose by 36–70% without compromising the diagnostic efficacy of the examination [32,55]. The continuous access to new data aggregated within networks will facilitate ongoing optimization of research protocols with minimal infrastructure modifications. AI algorithms will also play a pivotal role in scrutinizing patient data to forecast susceptibility to radiation-induced effects, thereby enabling more judicious decisions regarding the necessity and frequency of subsequent tomographic examinations. This comprehensive approach endeavors to augment the efficiency and safety of the diagnostic process while ensuring optimal patient care [56,57,58].

This study also presents several limitations that warrant acknowledgment. Firstly, it was conducted within a singular institution, potentially limiting the generalizability of the findings to settings with diverse equipment or protocols. Conducting multi-center studies would enhance the validation of these results and ensure their broader applicability. Secondly, the study’s conclusions are drawn from a retrospective analysis of 134 chest CT studies. Future research endeavors could benefit from an expanded and more diverse sample size, encompassing a broader spectrum of patient demographics and various types of CT scans. Thirdly, while the introduction of DLPss is innovative, its current form may harbor inherent limitations. It is imperative to investigate how variations in scanning parameters or patient conditions may influence its accuracy. Fourthly, the heterogeneity in CT scanners and technologies across different healthcare settings may influence the applicability of DLPs. Finally, the absence of long-term follow-up data in this study precludes an assessment of the clinical impact of DLPss in radiation dose estimation, particularly concerning patient outcomes or radiation-induced complications.

## 5. Conclusions and Future Scope

In summary, the findings of this study possess the potential to significantly influence clinical practice by advocating for the adoption of DLPss for radiation dose estimation, thereby signaling a departure from current standard practices towards the minimization of unnecessary radiation exposure and the optimization of CT scan protocols. To augment the applicability of these findings, future investigations should prioritize the expansion of the study population to encompass a larger and more diverse patient cohort, inclusive of various types of CT scans. Moreover, the implementation of multi-center trials will be indispensable in validating the efficacy and generalizability of DLPss across disparate clinical settings and technological platforms. Exploration into the seamless integration of DLPss into extant CT scanning software and protocols holds promise in enabling real-time, patient-specific dose optimization, thereby amplifying the safety and precision of medical imaging procedures. Additionally, conducting comparative studies juxtaposing DLPss with other emerging methodologies across diverse anatomical regions will furnish a more holistic comprehension of their respective strengths and limitations, thereby informing evidence-based clinical decision-making.

## Figures and Tables

**Figure 1 biomedicines-12-00600-f001:**
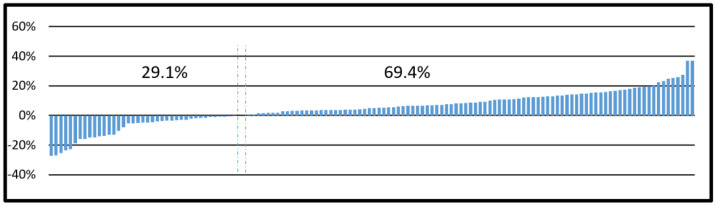
Percentage range of change in the length of the diagnostic scan compared to the topographic scan.

**Figure 2 biomedicines-12-00600-f002:**
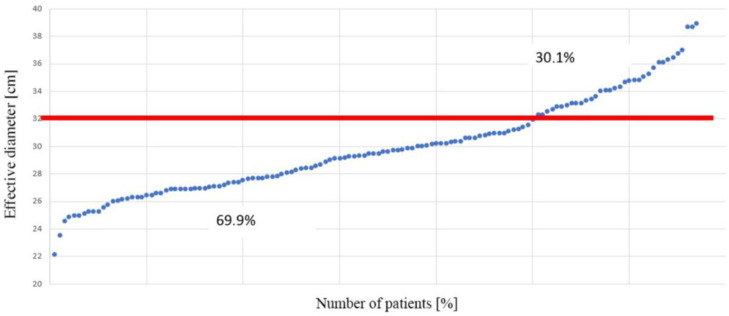
Percentage distribution of patients whose effective cross-sectional diameter in chest CT examination differed from the diameter of the reference phantom.

**Figure 3 biomedicines-12-00600-f003:**
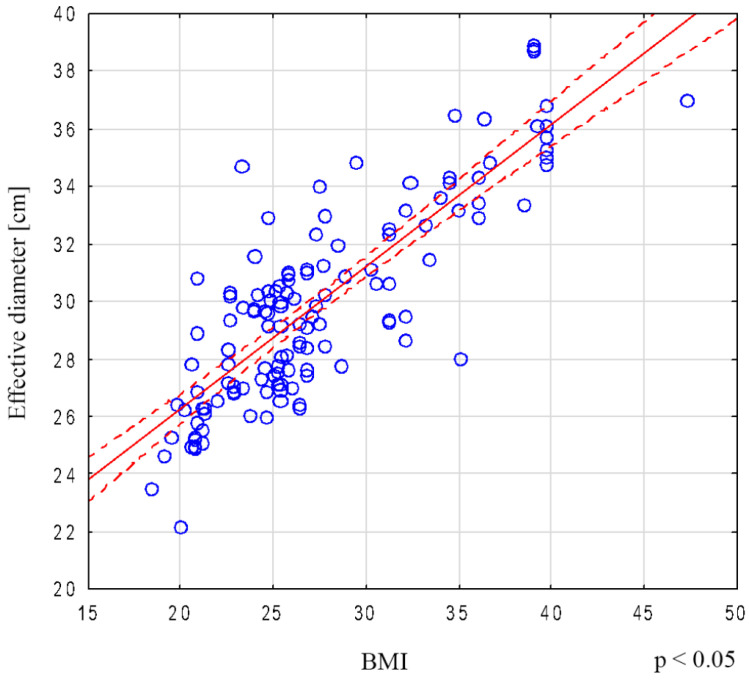
Correlation of BMI with effective chest diameter for the entire group.

**Figure 4 biomedicines-12-00600-f004:**
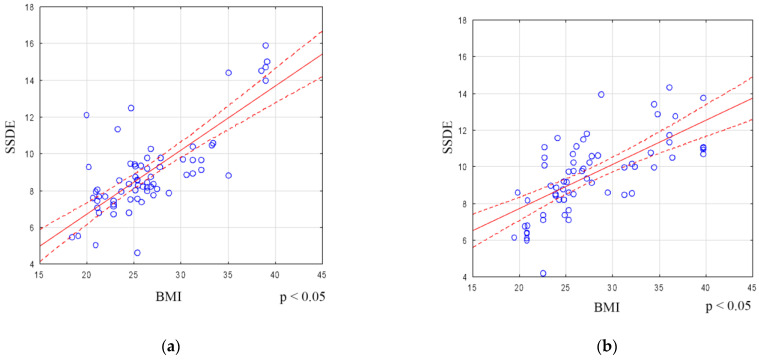
(**a**) Correlation of SSDE with BMI (female group), (**b**) correlation of SSDE with BMI (male group).

**Table 1 biomedicines-12-00600-t001:** Anthropometric parameters.

	Mean	Female	Male	*p*
No.	134	68	66	*p* = 0.9028
Age (y.o.)	64.54 ± 13.05	65.89 ± 12.91	63.15 ± 13.15	*p* = 0.14
Weight (kg)	77.89 ± 19.65	69.84 ± 15.98	86.18 ± 19.72	*p* < 0.05
Height (cm)	168.31 ± 9.87	161.04 ± 5.97	175.8 ± 7.10	*p* < 0.05
BMI	27.33 ± 5.81	26.88 ± 5.70	27.8 ± 5.93	*p* = 0.46

**Table 2 biomedicines-12-00600-t002:** Correlation of CTDIvol with selected anthropometric parameters (*p* < 0.05).

CTDIvol	Female	Male
Age	−0.08	−0.26
Weight	0.69	0.88
Height	−0.01	0.30
BMI	0.74	0.86
(LAT)	0.74	0.73
(AP)	0.72	0.79
Effective diameter (D_eff_)	0.79	0.87
Scan length	−0.43	−0.21

**Table 3 biomedicines-12-00600-t003:** Comparison of DLP [mGy·cm] values for the topogram and the diagnostic scans.

Type of Scan	Mean	Female	Male	*p*
topogram scan	5.28 ± 0.88	5.00 ± 0.86	5.58 ± 0.80	*p* < 0.05
diagnostic scan	286.51 ± 99.82	258.88 ± 98.97	314.98 ± 93.11	*p* < 0.05

## Data Availability

The datasets used and analyzed during the current study are available from the corresponding author on reasonable request.

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
