# Peer review of "Redefining Radiation Metrics: Evaluating Actual Doses in Computed Tomography Scans"

_biomedicines, 2024, doi:10.3390/biomedicines12030600_

Round 1

Reviewer 1 Report

Comments and Suggestions for Authors

There are several queries raised based on careful examination of the manuscript. My comments are as follows:

1. Title must be changed. It should appear as a scientific article.

2. What is the author's specific contribution as compared to published literature. Kindly add necessary details.

3. The relevance and explanation of Fig.1. and Fig.2 Should be included in revised version.

4. The utility of outcomes is not clear to the readers. How this study benefits and is adding significant insight need to be carefully included.

5. Limitations and future scope of proposed methodology need to be carefully written.

Comments on the Quality of English Language

English Language is OK.

Reviewer 2 Report

Comments and Suggestions for Authors

Authors should specify in the method the type of the study  and also the type of CT examined if total body, HRCT, abdominal ct, skull ct.. This information is necessary.  Authors should discuss if artificial intelligence can have  a role in the radioprotection in the future

Reviewer 3 Report

Comments and Suggestions for Authors

Introduction

- The introduction commendably establishes the significance of CT scanning in the medical field and its associated radiation risks, highlighting the necessity for optimised radiation parameters. However, it could be improved by providing a more detailed review of previous studies, thus creating a stronger foundation for the research gap the study aims to address. A clearer articulation of the study's objectives, in relation to existing knowledge, would enhance its impact and relevance.

Materials and Methods

- This section is well-structured, detailing the methodology with a focus on the retrospective analysis of CT scans and the development of the DLPss parameter. However, the specifics of the patient selection process and data collection methods are somewhat underdeveloped. Elaborating on these aspects would improve replicability and scientific rigour. Additionally, a more detailed explanation of statistical tools and techniques used would benefit readers less familiar with the field.

Results

- The presentation of results is clear, with effective use of statistical analysis and visual aids. However, the integration of these results with broader scientific concepts is somewhat limited. Expanding on how these results compare with existing literature, and what new insights they offer, would greatly enhance the section. Furthermore, discussing the potential clinical implications of these findings in a more detailed manner would provide greater depth.

Discussion

- The discussion skillfully connects the study's findings with the broader context of radiological practice and radiation safety. However, it could benefit from a more thorough exploration of the limitations of the study and potential areas for future research. A critical analysis of the study's methodology and its impact on the results would provide a more balanced view. Additionally, discussing potential applications and implications for clinical practice would add value.

Comments on the Quality of English Language

Overall, the manuscript is well-written, with a professional and academic tone. The language is mostly clear and concise, making the content accessible. However, there are occasional grammatical errors and instances of awkward phrasing that could be improved with careful editing. Ensuring consistency in terminology and refining sentence structure would enhance the readability and professional appearance of the manuscript.

Round 2

Reviewer 2 Report

Comments and Suggestions for Authors

 The authors made major corrections and the manuscript quality is now improved

Reviewer 3 Report

Comments and Suggestions for Authors

satisfied with changes